# Conformational Preferences of Pyridone Adenine Dinucleotides from Molecular Dynamics Simulations

**DOI:** 10.3390/ijms231911866

**Published:** 2022-10-06

**Authors:** David P. Buckley, Marie E. Migaud, John J. Tanner

**Affiliations:** 1Department of Biochemistry, University of Missouri, Columbia, MO 65211, USA; 2Mitchell Cancer Institute, Department of Pharmacology, College of Medicine, University of South Alabama, Mobile, AL 36604, USA; 3Department of Chemistry, University of Missouri, Columbia, MO 65211, USA

**Keywords:** nicotinamide adenine dinucleotide, molecular dynamics simulations, oxidative stress

## Abstract

Pyridone adenine dinucleotides (ox-NADs) are redox inactive derivatives of the enzyme cofactor and substrate nicotinamide adenine dinucleotide (NAD) that have a carbonyl group at the C2, C4, or C6 positions of the nicotinamide ring. These aberrant cofactor analogs accumulate in cells under stress and are potential inhibitors of enzymes that use NAD(H). We studied the conformational landscape of ox-NADs in solution using molecular dynamics simulations. Compared to NAD^+^ and NADH, 2-ox-NAD and 4-ox-NAD have an enhanced propensity for adopting the *anti* conformation of the pyridone ribose group, whereas 6-ox-NAD exhibits greater *syn* potential. Consequently, 2-ox-NAD and 4-ox-NAD have increased preference for folding into compact conformations, whereas 6-ox-NAD is more extended. ox-NADs have distinctive preferences for the orientation of the pyridone amide group, which are driven by intramolecular hydrogen bonding and steric interactions. These conformational preferences are compared to those of protein-bound NAD(H). Our results may help in identifying enzymes targeted by ox-NADs.

## 1. Introduction

Nicotinamide adenine dinucleotide and nicotinamide adenine dinucleotide phosphate (NAD and NADP, respectively) are cofactors and substrates for numerous enzymes and have a multitude of redox and non-redox roles in cells [1,2]. NAD(P)(H) plays an essential role in oxidoreductase catalyzed reactions, where the C4 of the nicotinamide ring (Figure 1A) is the direct hydride acceptor for dehydrogenases or the hydride donor for reductases. NAD(P)H is also the hydride donor to the flavin cofactors, FAD and FMN, in a large class of enzymes known as flavin-dependent monooxygenases, which catalyze the addition of O atoms from O_2_ to substrates [3]. The Sir2 family of deacetylases use NAD^+^ as a substrate to catalyze the deacetylation of acyllysine residues of protein substrates to reveal the free amino form of the lysine sidechain and 2′-*O*-acyl-ADP-ribose [4]. ADP-ribosyltransferases use NAD^+^ as the source of an adenosine diphosphoriboside (ADP-ribose) unit to covalently link single or multiple ADP-ribose units to protein substrates [5], while glycohydrolases are responsible for the loss of the nicotinamide unit and formation of ADP-ribose. The ubiquity of NAD(P) in the metabolism is reflected in the Protein Data Bank, where over 4000 entries contain NAD^+^, NADH, NADP^+^, or NADPH complexed to the protein.

The structural properties of NAD(H) have been measured by a variety of experimental techniques, including solution NMR and fluorescence, and X-ray crystallography. Early NMR studies have indicated that NAD(H) adopts a folded conformation in solution that is in rapid equilibrium with an open form [6,7,8]. Measurements of fluorescence anisotropy decay have indicated NADH has an apparent hydrodynamic radius in solution of about 6 Å [9]. Estimates of the hydrodynamic volume of NADH in aqueous solution are in the range of 500–600 Å^3^ [10,11]. The relative population of folded NADH in aqueous solution is estimated to be 25–35% [6,9]. Another NMR study suggested NAD^+^ is 15% folded in solution [12]. A crystal structure of the Li^+^ salt of NAD^+^ shows an extended molecule with an inter-base distance of 12 Å [13,14]. The crystal structure shows a dimer in which the adenine of one molecule is stacked intermolecularly on the nicotinamide of a neighboring molecule.

Pyridone adenine dinucleotides (ox-NAD(P)) are oxidized derivatives of NAD(P) possessing a carbonyl group on the nicotinamide ring at the C_2_^N^, C_4_^N^, or C_6_^N^ position (Figure 1B). These aberrant forms of NAD(P) accumulate in cells under stress conditions when cells build up an overabundance of electrons, in turn causing an abnormally high ratio of NADH to NAD^+^ and the generation of reactive oxygen species [15,16,17]. Pyridone adenine dinucleotides can be detected at concentrations within the nmol/mg of protein range of NAD(P)(H) concentrations in tissues and cultured cells [18,19]. There, they have the potential to inhibit NAD(P)-dependent enzymes, causing further metabolic dysfunction and pathogenic oxidative damage.

The association of pyridone adenine dinucleotides and their catabolites in age-related diseases, including cancer and cancer metastasis [19], and the potential for their inhibition of metabolic enzymes motivated us to study the in-solution conformations of the three major pyridone adenine dinucleotides. Herein, we report molecular dynamics simulations of NAD^+^, NADH, 2-ox-NAD, 4-ox-NAD, and 6-ox-NAD (Figure 1). The presence of a carbonyl group on the nicotinamide ring profoundly alters the conformational landscape compared to NAD(H), most notably the compactness of the dinucleotide and the preferences of the pyridone ribose glycosidic bond dihedral angle and pyridone amide group orientation. We also compare the conformations of ox-NADs in solution with enzyme-bound NAD(H) to assess the compatibility of these cofactor mimics with enzyme active sites. Our results may be useful for identifying enzymes targeted by ox-NADs.

## 2. Results

### 2.1. Conformational Preferences of NAD^+^ and NADH

Ten 100 ns MD simulations (1 µs total) for both NAD^+^ and NADH (Figure 1A,B) were performed as references for the simulations of ox-NADs. The spatial extent of NAD(H) was assessed by monitoring the distance between the centroids of the nicotinamide and adenine rings. The time evolution of this parameter reveals that NAD(H) underwent multiple opening/closing events during the simulation (Appendix A). The inter-base distance varied from a minimum of 3.2 Å to a maximum of 17.2 Å for NAD^+^, and a minimum of 3.4 Å to a maximum of 17.9 Å for NADH. The distribution of this parameter was trimodal for both NAD^+^ and NADH, with peaks near <6 Å, 8–10 Å, and 12–15 Å, which we refer to as folded, semi-extended, and extended, respectively (Figure 2A,B). The folded peak was much more prominent for NADH, suggesting a greater propensity for adopting compact conformations in solution.

Folded NAD(H) tends to have bases stacked in near-parallel arrangements. This feature is evident in the scatter plot of inter-base distance versus the normal–normal inter-base plane angle (Figure 3A,B). For reference, parallel base stacking corresponds to normal–normal plane angles of 0° and 180°. The most common folded NAD^+^ conformation (Figure 4A) was identified as the event where the distance between atoms O_4_′^N^-C_4_
^A^ was <=4.0 Å. Rarely, the most common folded NAD^+^ transitioned to a highly base-stacked conformation (Figure 4B). This conformation corresponds to the small population with inter-base distances less than 4 Å and an inter-base angle of 160–180° (upper left of Figure 3A). Compared to NAD^+^, NADH displayed a stronger population density in its folded regions, which is characterized by two clusters with inter-base angles of 0–45° or 135–180° (Figure 3B). Examples of these folded NADH conformations are shown in Figure 4E (inter-base angle of 30°) and Appendix A (inter-base angle of 158°). Examples of semi-extended and extended conformations of NAD^+^ and NADH are provided in Figure 4C,D and Figure 4F,G, respectively.

The conformational space of the nicotinamide riboside group of NAD(H) is of particular interest for comparison to ox-NADs. The conformation of the nicotinamide riboside was assessed by monitoring the *N*-glycosidic bond dihedral angle (χ_N_ in Table 1) and the amide dihedral angle (θ in Table 1). NAD(H) showed a preference for χ_N_ in the range of −160° ± ~70° (Figure 5A,B), which corresponds to the *anti* conformation of the base [20]. In the *anti* conformation, the base is rotated away from the sugar, as shown in Figure 4A for NAD^+^ and Figure 4E for NADH. A somewhat smaller population with χ_N_ of 0–45° corresponded to the *syn* conformation. In the *syn* conformation, the base is rotated about the glycosidic bond to occupy a space closer to the sugar ring (examples of *syn* NAD^+^/NADH are provided in Appendix A, respectively). The preference for *anti* reflects a lower steric clash between the nicotinamide base and ribose compared to *syn*. The data suggest slightly greater *syn* occurrence for NADH compared to NAD^+^ (Figure 5A,B).

The amide angle (θ) for NAD(H) also shows very distinct preferences. Two large populations near 0° and 180° are observed for NAD^+^ (Figure 5B), whereas NADH strongly prefers θ near 0° (Figure 5B). Poses with θ near 0° are shown in Figure 4.

Finally, the adenosine in NAD(H) strongly preferred *anti* over *syn*. This is expected for a purine base due to its larger size compared to pyrimidines (Appendix A).

### 2.2. Foldedness of ox-NADs

MD simulations of ox-NADs were performed to understand how the presence of a carbonyl group at the C_2_^N^, C_4_^N^, or C_6_^N^ of the nicotinamide (Figure 1B) affects the conformational landscape of the dinucleotides. The inter-base distance was monitored to determine how the oxidation of the nicotinamide affects dinucleotide foldedness. Similar to NAD^+^ and NADH, the distribution of inter-base distance for ox-NADs is trimodal; however, the relative populations of folded, semi-extended, and extended differ dramatically from those in NAD(H) (Figure 2). For example, the inter-base distance distributions for 2-ox-NAD and 4-ox-NAD show a major peak at 4–5 Å, indicating a greater tendency to fold compared to NAD^+^ or NADH (Figure 2). This increase in the folded population comes at the expense of fewer extended conformations. Interestingly, the peak representing folded 2-ox-NAD and 4-ox-NAD is shifted by −1 Å compared to NAD^+^, indicating greater compactness and likeness to NADH. Similar to NADH, the folded conformations of ox-NADs exhibit base stacking, as indicated by the dense populations with inter-base distance <5 Å and inter-base plane angle near 30° and 160° (Figure 3C,D).

Oxidation at the nicotinamide C6 results in significant variability in folding potential. Interestingly, individual plots of the inter-base distance distribution for each simulation of 6-ox-NAD highlight one simulation, S1, where the number of folded conformations of 6-ox-NAD reached nearly 12,000, far more than in any other of the other simulations (Appendix A). We attribute this event to an extended specific interaction with a neutralizing Na^+^ ion. This association appears to further stabilize a folded form observed in 6-ox-NAD (Figure 6G), which exists at other times of the trajectory uninfluenced by Na^+^. Additionally, this fold pattern was observed in the other ox-NADs (not shown) as well as in NADH (Appendix A). While this individual simulation likely skews the standard deviation for 6-ox-NAD in bins of the folded region, we note that this strong Na^+^ interaction was identified in only one out of the fifty simulations performed and appears to be a rare event. On average, the distribution of inter-base distance of 6-ox-NAD shows a predominant peak at 13–14 Å, representing extended conformations, while the populations of folded and semi-extended conformations are relatively minor when compared to 2- and 4-ox-NAD (Figure 2E). Thus, on average, 6-ox-NAD is more extended than the other ox-NADs.

### 2.3. Pyridone Ribose Conformation of ox-NADs

Oxidation of the nicotinamide at C_2_^N^ and C_6_^N^ profoundly affects the conformation of the N-glycosidic bond dihedral angle. 2-ox-NAD shows a very strong preference for the *anti* conformation (χ_N_ ~−135° to −180°) and near complete avoidance of *syn* (Figure 5C). This tendency likely reflects the steric clash that arises in the *syn* conformation between the C_2_^N^-carbonyl and the ribose. The distribution of χ_N_ for 4-ox-NAD resembles that of NAD(H) in that both *anti* and *syn* can be observed, with a preference for the former (Figure 5D). Oxidation at the C_6_^N^ has the opposite effect: 6-ox-NAD adopts *syn* far more than any of the dinucleotides (Figure 5E). The 6-ox-NAD pyridone ribose likely experiences steric clash in either the *anti* or *syn* conformations, as both forms position atoms above the ribose. The poses in Figure 6 demonstrate *anti* conformations of the pyridone ribose of ox-NADs.

In addition to these large trends in χ_N_ preferences, ox-NADs also exhibit more subtle deviations from the conformational tendencies of NAD(H). For example, 2-ox-NAD has a small population near χ_N_ ~−60°, which is not present in the other dinucleotides (Figure 5C). An example of this conformation is shown in Appendix A. Note the approach of the C_2_^N^-carbonyl oxygen to the ribose oxygen O_4_′^N^. This small population of χ_N_ ~−60° likely represents the *high-anti* pyridone ribose conformation, which corresponds to the closest approach of the C_2_^N^-carbonyl oxygen to the ribose O_4_′^N^ atom [21,22]. Additionally, 6-ox-NAD exhibits a small population near χ_N_ ~110° (Figure 5E). An example of this conformation is shown in Appendix A. Similar to 2-ox-NAD, the C_6_^N^-carbonyl oxygen of 6-ox-NAD is positioned near the ribose oxygen O_4_′^N^. The population for χ_N_ ~110° likely represents the *high-syn* pyridone ribose conformation. As 6-ox-NAD readily adopts both *anti* and *syn* conformations for its pyridone ribose, the *high-syn* population could be a transition state between the two main conformations of the pyridone ribose glycosidic bond angle. Conversely, 4-ox-NAD appears to sample the *high-anti* and *high-syn* regions to a very limited extent compared to 2- and 6-ox-NAD (Figure 5D).

Ox-NADs also have distinct preferences for the amide dihedral angle (θ). 2-ox-NAD exhibits three major populations centered at θ of 180°, +45°, and −45° (Figure 5C). Conformations with θ near ±180° enjoy hydrogen bonding between the C_2_^N^ carbonyl and the amide N_7_^N^ (Figure 6A,B). 2-ox-NAD avoids θ = 0°; at θ = 0° the C_2_^N^- and amide carbonyls would clash. Instead, 2-ox-NAD samples θ angles near ±45° in which the carbonyls are offset (Figure 6C). Maximizing hydrogen bonding while avoiding clashing carbonyls also explains the θ preferences of 4-ox-NAD. 4-ox-NAD favors θ = 0° (Figure 5D), which enables hydrogen bonding between the C_4_^N^ carbonyl and the amide N_7_^N^ (Figure 6D and Figure 7F). 4-ox-NAD avoids θ = 180°, which would result in carbonyls clashing, and instead exhibits offset conformations of near θ of ±160° (Figure 5D and Figure 6E). 6-ox-NAD exhibits approximately equal preferences for θ of 0° and 180° with a *syn* pyridone ribose (Figure 5E and Appendix A), and a preference for θ centered at 0° with an *anti* pyridone ribose (Figure 5E and Figure 6G).

### 2.4. Puckering of the Pyridone Ribose

The puckering of the pyridone ribose was examined by monitoring the v_2_ angle (Table 1). NAD^+^ strongly prefers C2′-endo (v_2_ near −35 to −40° [20]), especially when the nicotinamide is *syn* (Figure 7A). In contrast, NADH prefers C3′-endo (v_2_ ~35–40°) independent of the nicotinamide glycosidic bond angle (Figure 7B). 4-ox-NAD and 6-ox-NAD exhibit a strong preference for C3′-endo, whereas 2-ox-NAD samples both pucker conformations. 6-ox-NAD is unique in that it rarely adopts the C2′-endo conformation, but experiences near “flat” ribose conformations (v_2_ ~0°) with the pyridone ribose *anti*, perhaps as a result of the C_6_^N^-carbonyl oxygen contacting the ribose.

### 2.5. Conformations of Protein-Bound NAD(H)

The conformational tendencies of NAD(H) bound to proteins were studied by mining the PDB. The dataset consisted of 3965 poses of NAD^+^ and 671 poses of NADH in crystal structures with resolution of 2.9 Å or better. The trends for NAD^+^ and NADH were found to be similar; hence, we show results for the combined dataset of 4636 poses.

NAD(H) tends to adopt highly extended conformations when bound to proteins (Figure 8A). The average centroid–centroid distance between the bases is 14.0 Å with standard deviation of 1.3 Å. The distribution is similar to those reported previously [23,24]. Note the average of 14.0 Å is close to the typical inter-base distance of extended conformations of NAD(H) and ox-NADs in solution (Figure 2).

The glycosidic bond dihedral angle distribution of protein-bound NAD(H) resembles that of NAD(H) in solution. The adenosine is found almost exclusively in the *anti* conformation (Figure 8B), whereas the nicotinamide ribose adopts both the *anti* and *syn* conformations (Figure 8C).

The amide dihedral angle (θ) of protein-bound NAD(H) is near 0° or 180°, with a preference for the latter (Figure 8C). This contrasts the behavior of NAD(H) in solution, which strongly prefers θ near 0° (Figure 5A,B). The discrepancy between solution and protein-bound NAD(H) likely reflects the fact that hydrogen bonding with the protein tends to establish the orientation of the amide group of protein-bound NAD(H). An example is aldehyde dehydrogenase 4A1, where a hydrogen bond with a backbone carbonyl fixes the amide group at ~180° [25,26].

## 3. Discussion

We compared the in-solution conformations of NAD^+^, NADH, and ox-NADs using MD. The results for NAD^+^ differ from those of a much shorter, 5 ns simulation of NAD^+^ performed by us over 20 years ago [27]. In that simulation, only a single opening and closing event was observed, and NAD^+^ spent approximately 80% of the time in compact conformations with the inter-base distance near 5 Å. In contrast, the new multiple 100 ns simulations showed many opening and closing events and a preference for open conformations, with the inter-based distance in the range of 8–14 Å. We attribute this discrepancy to two factors: the short timescale of the previous simulation limiting the sampling of the conformational space of NAD^+^, and the uniqueness of the starting conformation of NAD^+^ used for simulation from PDB 2BKJ, chain B. In that structure, NAD^+^ is tightly folded with a *syn* adenine, which was not observed in our current simulation (Appendix A). Overlaying what we found to be the rarest folded form of NAD^+^ (Figure 4B) from our current simulations with the starting conformation from PDB 2BKJ, chain B shows an interesting agreement between the folds overall, aside from the flipped adenine orientation (Appendix A). We note that in the current simulations we observed compact conformations persisting for ~5 ns, i.e., the entire duration of the previous simulation (see near t = 80, t = 200, or t = 700 ns in Appendix A).

We suggest the current simulation more accurately depicts the behavior of NAD^+^ in solution because it more thoroughly sampled the available phase space compared to the previous simulation. In particular, we observed numerous opening and closing events for all five dinucleotides, suggesting that the sampling of pyrophosphate dihedral angles is adequate (Appendix A). The consistency of the end-to-end distributions calculated from the individual 100 ns simulations also implies good sampling of these angles (Appendix A). The average radius of the gyration of NADH from the simulation of 5.2 ± 0.6 Å is consistent with the experimental value of 6 Å [9]. Further, the glycosidic bond angles of both bases sampled both *anti* and *syn* conformations, indicating they have not been trapped in local minima. An exception is χ_N_ of 2-ox-NAD, but this can be rationalized by the strong steric clash of the C_2_^N^-carbonyl with the ribose inhibiting the formation of *syn*. Finally, rotations around θ were observed for all five dinucleotides, indicating that this degree of freedom was also sampled adequately.

The simulations used force fields built by the chemical similarity to groups of known parameterizations using the CHARMM General Force Field generator, an approach used widely in simulations of protein complexes with organic molecules. The force fields for ox-NADs draw heavily from those developed previously by MacKerell’s group for NAD(H) [28]. Although higher level force field development for ox-NADs using quantum mechanical methods is possible, we consider the current approach to be adequate given our main goal of understanding the tendencies of dihedral angles. Indeed, the preferences obtained from the simulations are in line with expectations, such as the adenosine of ox-NADs preferring *anti*, and the pyridone ribose glycosidic bond reflecting the potential for a steric clash between the carbonyl at C_2_^N^, C_4_^N^, or C_6_^N^ and the ribose. These results suggest that the simulations contain useful information.

Ox-NADs exhibit distinct preferences for the conformation of the pyridone ribose. 2-ox-NAD strongly prefers the *anti* conformation, whereas 6-ox-NAD prefers *syn*. These tendencies are driven by the avoidance of steric clashes between the added carbonyl and the ribose. Ox-NADs also have preferences for the orientation of the pyridone amide group, which are driven by hydrogen bonding and steric interactions with the carbonyl at C_2_^N^, C_4_^N^, or C_6_^N^.

The conformational preferences of the pyridone ribose appear to influence the folding of the dinucleotides. On average, 2-ox-NAD and 4-ox-NAD exhibit enhanced folding compared to NAD(H), whereas 6-ox-NAD is more extended than the other dinucleotides. We suggest these trends reflect the unique χ_N_ preferences of the dinucleotides, as shown in the scatter plot of inter-base distance versus χ_N_ (Figure 9). The plots show that for all five dinucleotides, the highly compact conformations (i.e., distance < 5 Å) occur most frequently when the nicotinamide/pyridone ribose is *anti*, i.e., χ_N_ in the range of 135° to −180°. 2-ox-NAD and 4-ox-NAD prefer the *anti* conformation of pyridone ribose, leading to enhanced folding. Conversely, 6-ox-NAD shows substantial syn conformation, leading to less folding.

Ox-NADs have the potential to bind to enzymes and thus act as inhibitors. The comparison of the conformational preferences of ox-NADs in solution to those of protein-bound NAD(H) provides insight into proteins potentially targeted by ox-NADs. For example, because of its strong preference for χ_N_
*anti*, 2-ox-NAD may not be accommodated in the active sites of enzymes that bind NAD(H) with the nicotinamide *syn*. Conversely, 6-ox-NAD may have poor complementarity with enzymes that bind NAD(H) with the nicotinamide *anti*. NAD(H) bound to proteins strongly prefers extended conformations, and we expect that ox-NADs will bind proteins in similarly extended conformations. For this reason, there may be an energetic cost associated with unfolding for ox-NADs, particularly 2-ox-NAD and 4-ox-NAD, which seem to prefer folded conformations over 6-ox-NAD. This information may be useful for prioritizing experiments on the inhibition of enzymes by ox-NADs.

## 4. Computational Methods

### 4.1. Molecular Dynamics (MD) Simulations of NAD^+^, NADH, and ox-NADs

The starting coordinates for the molecular dynamics (MD) simulations of NAD^+^ and NADH were obtained from structures of an aldehyde dehydrogenase enzyme (NAD^+^: PDB 5KF6, chain A; NADH: PDB 7MYC, chain A). In these structures, NAD(H) adopts the classical extended conformation observed for Rossmann fold domains. The molecular editor Avogadro v1.2.0 [29] was used to generate starting coordinates for the three ox-NAD simulations by adding a carbonyl functional group at C_2_^N^, C_4_^N^, or C_6_^N^ of the nicotinamide ring, hence eliminating the +1 formal charge on N_1_^N^ from the starting coordinates for NAD^+^ (Figure 1). The pyrophosphate of the dinucleotide was assumed to have a charge of −2, appropriate for pH 7.0. File formatting and topology generation for each of the dinucleotides was performed as described by [30]. The CHARMM General Force Field (CGenFF version 4.6—July 2021) [31,32] was used for toppar stream file creation, and the conversion of each stream file from CHARMM to GROMACS format was performed with the Python v2.7 [33] CGenFF program, using NetworkX v1.11 [34]. The force fields for ox-NADs are based on those developed previously by MacKerell’s group for NAD(H) [28].

The following procedure was used to generate MD trajectories for each of the five dinucleotides (NAD^+^, NADH, 2-ox-NAD, 4-ox-NAD, 6-ox-NAD). The GROMACS 2018.3 package was used for solvation of the system, equilibration, and production [35,36,37,38,39]. The dinucleotide was solvated with the TIP3P water model [40] in a rhombic dodecahedral box of 35–41 nm^3^ and neutralized with Na^+^ ions (one Na^+^ for NAD^+^, two Na^+^ for NADH and ox-NADs). The solvated system was energy-minimized using the steepest descent algorithm until the maximum force was less than 10.0 kJ mol^−1^ nm^−1^. A second energy minimization was performed with the inclusion of van der Waals forces using a switching function with inner and outer cutoffs of 1.0 nm and 1.2 nm, respectively. The resultant energy-minimized structure was positionally restrained by its non-hydrogen atoms and the system was subjected to a 100 ps equilibration under NVT (constant particle number, volume, and temperature) conditions at 300 K with the v-rescale thermostat [41] and using a time constant of 0.1 ps. The system was then further equilibrated for 100 ps using NPT (constant particle number, pressure, and temperature) conditions and isotropic Berendsen pressure coupling [42] with the reference pressure equal to 1 bar, isothermal compressibility set to 4.5 × 10^−5^ bar^−1^, and time constant set to 2 ps. Ten independent 100 ns production simulations were performed after release of positional restraints on the starting NPT-equilibrated structure. The Particle Mesh Ewald method (real space cutoff of 1.2 nm) was used for the calculation of electrostatic forces [43,44], and isotropic Parrinello-Rahman pressure coupling [45,46,47] was implemented with the same reference pressure and isothermal compressibility as in NPT equilibration. Periodic boundary conditions were applied in all three directions. All MD simulations were performed using a 2 fs time steps. Coordinates were saved every 1 ps (every 500 steps) for analysis. The details of the simulations are listed in Table 2.

### 4.2. Analysis of Conformations

Analyses of the MD trajectories were performed using the MDAnalysis package v2.0.0 [48,49]. For each dinucleotide, the ten 100 ns simulations were concatenated into a single, time-contiguous 1 µs trajectory. The end-to-end distance of a dinucleotide was defined as the distance between the centroids of the nicotinamide and adenine rings (i.e., the inter-base centroid–centroid distance). Dihedral angles were defined as in [24] (Table 2). The angle between the nicotinamide and adenine rings was calculated using normal vectors, as follows. Two vectors in the plane of the nicotinamide ring were defined as the vector from atom N_1_^N^ to C_2_^N^ (vector ***a***) and from N_1_^N^ to C_6_^N^ (vector ***b***). The vector normal to the nicotinamide ring was calculated as ***a*** × ***b***. Two vectors in the plane of the adenine were defined as the vector from atom C_5_^A^ to C_4_^A^ (vector ***c***) and C_5_^A^ to C_6_^A^ (vector ***d***). The vector normal to the adenine was calculated as ***c*** × ***d***. The normal–normal angle between the two bases was calculated as the inverse cosine of the dot product between the two normal vectors. Graphs were created with Origin 2021 v9.85.204. Tracking folded conformations for NADH/ox-NADs was performed by tallying the number of instances in which the distance between the following atoms was less than or equal to 3.3 Å, indicating a potential intramolecular hydrogen bond: [O_3_′^N^-O_2_′^A^], [O_3_′^A^-O_1_^N^], [O_3_′^A^/O_2_′^A^-O_4_′^N^] and [O_3_′^N^/O_2_′^N^-O_4_′^A^]. The individual trajectory frames selected for the visualization of folded conformations were from the most prevalent of the four bins described above. The individual trajectory frames selected for the visualization of semi-extended and extended conformations were from frames immediately preceding the folded frames obtained as described above, in which the inter-base centroid–centroid distance was ±0.5 Å from the respective molecule’s peak maximum in the distribution of inter-base distance (~9.0 Å for semi-extended, ~13 Å for extended).

Analysis of conformational clusters was performed as follows. For each dinucleotide, the RMSD was calculated across the concatenated 1 μs simulation using a reference conformation. Clusters of conformations were obtained by selecting frames of the trajectory where the RMSD to the reference was less than or equal to 1.0 Å. Ten frames that satisfied the RMSD criterion were selected for visualization, such that the frames were >5 ns in simulation time apart from each other, except for extended conformations of 4/6-ox-NAD, which had relatively few instances.

### 4.3. PDB Data Mining

The conformations of NAD^+^ and NADH bound to proteins were studied by mining the Protein Data Bank (PDB). A dataset of NAD(H) conformations was assembled by downloading X-ray crystal structures (asymmetric unit) with a resolution of 2.9 Å or better and containing either NAD^+^ (chemical ID NAD) or NADH (chemical ID NAI) noncovalently bound to the protein. PDB entries having only C-alpha models of the protein were omitted. The resulting dataset consisted of 1445 entries with NAD^+^ and 248 entries with NADH, and included 3965 instances of NAD^+^ and 671 of NADH. The conformations of the dinucleotides were analyzed using CNS version 1.3 [50].

## 5. Conclusions

We investigated the structure and dynamics of NAD^+^, NADH, and ox-NADs in solution using MD. All five dinucleotides exhibit a rapid equilibrium of folded, semi-extended, and extended conformations. Folded conformations are characterized by inter-base distances of less than 6 Å and the near-parallel stacking of the bases, whereas in typical extended conformations, the bases are separated by 13–14 Å. Among the ox-NADs, 2-ox-NAD and 4-ox-NAD exhibit a greater inclination for folding than 6-ox-NAD. This trend likely reflects the strong preference of 2-ox-NAD and 4-ox-NAD to adopt *anti* conformations of the pyridone ribose, whereas 6-ox-NAD shows an opposite preference for *syn*. The higher *anti* character of 2-ox-NAD and 4-ox-NAD leads to more facile base stacking and hence a higher probability of folding. The preference of 2-ox-NAD for the *anti* conformation of the pyridone ribose is especially strong and reflects the steric clash between the pyridone carbonyl and ribose that occurs in the *syn* conformation. The conformations of protein-bound NAD(H) were investigated by mining the PDB. Unlike NAD(H) in solution, the dinucleotide is almost exclusively in an extended conformation when bound to proteins. The nicotinamide ribose of protein-bound NAD(H) can be found in *anti* or *syn* glycosidic bond conformations, which reflects the unique interactions in the active site and specific stereochemical requirements of catalysis. The comparison of the conformational tendencies of ox-NADs in solution with those of protein-bound NAD(H) may aid in the identification of enzymes that have the potential to be inhibited by ox-NADs.

## Figures and Tables

**Figure 1 ijms-23-11866-f001:**
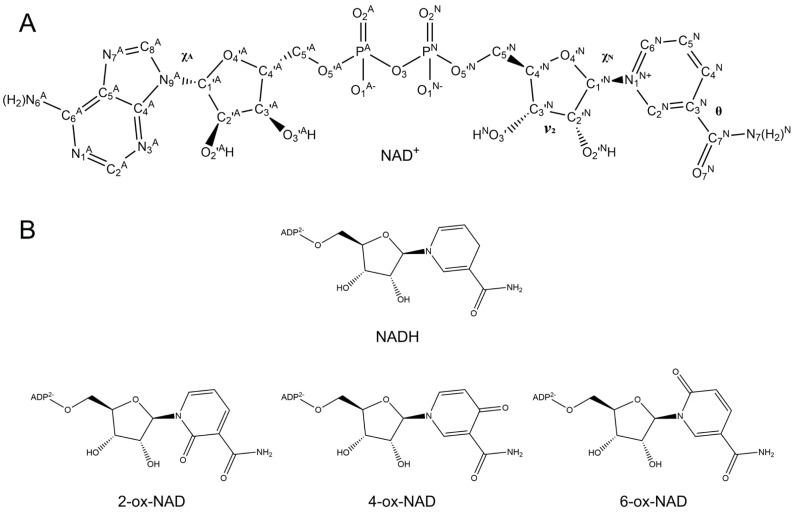
Chemical structures of NAD^+^, NADH, and ox-NADs. (**A**) NAD^+^ with atom labels indicated. Selected dihedral angles are noted. (**B**) Structures of NADH and ox-NADs. Adenosine diphosphate is simplified as ADP^2−^.

**Figure 2 ijms-23-11866-f002:**
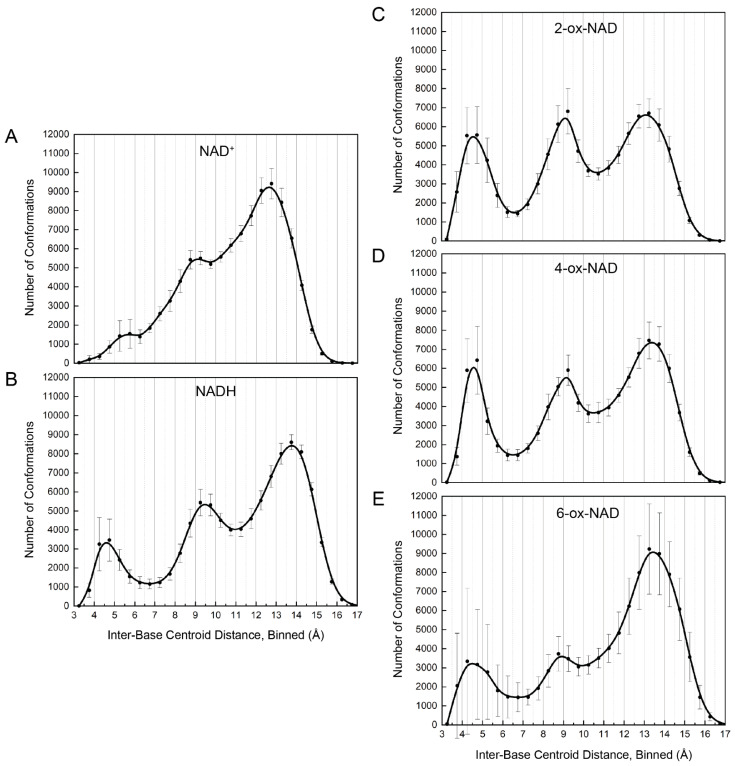
Average distribution and standard deviation of the inter-base centroid–centroid distance between the nicotinamide and adenine bases for ten 100 ns simulations of (**A**) NAD^+^, (**B**) NADH, (**C**) 2-ox-NAD, (**D**) 4-ox-NAD, and (**E**) 6-ox-NAD. Individual bin values from the average of the ten simulations are plotted as dots to indicate the center of the standard deviation. The distributions from the individual 100 ns simulations are shown in Appendix A.

**Figure 3 ijms-23-11866-f003:**
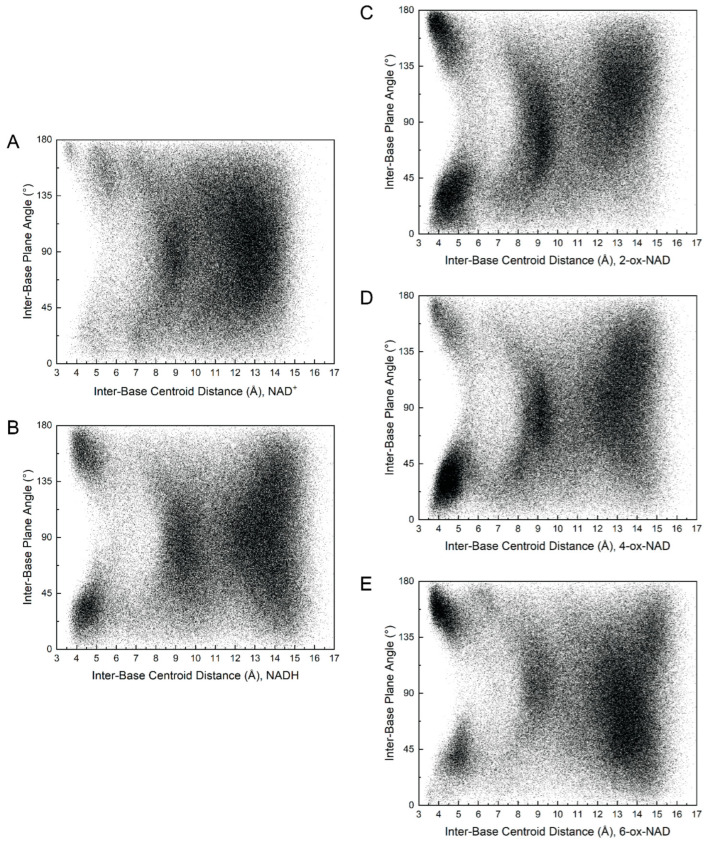
Scatter plot of the inter-base plane angle versus inter-base centroid–centroid distance for (**A**) NAD^+^, (**B**) NADH, (**C**) 2-ox-NAD, (**D**) 4-ox-NAD, and (**E**) 6-ox-NAD.

**Figure 4 ijms-23-11866-f004:**
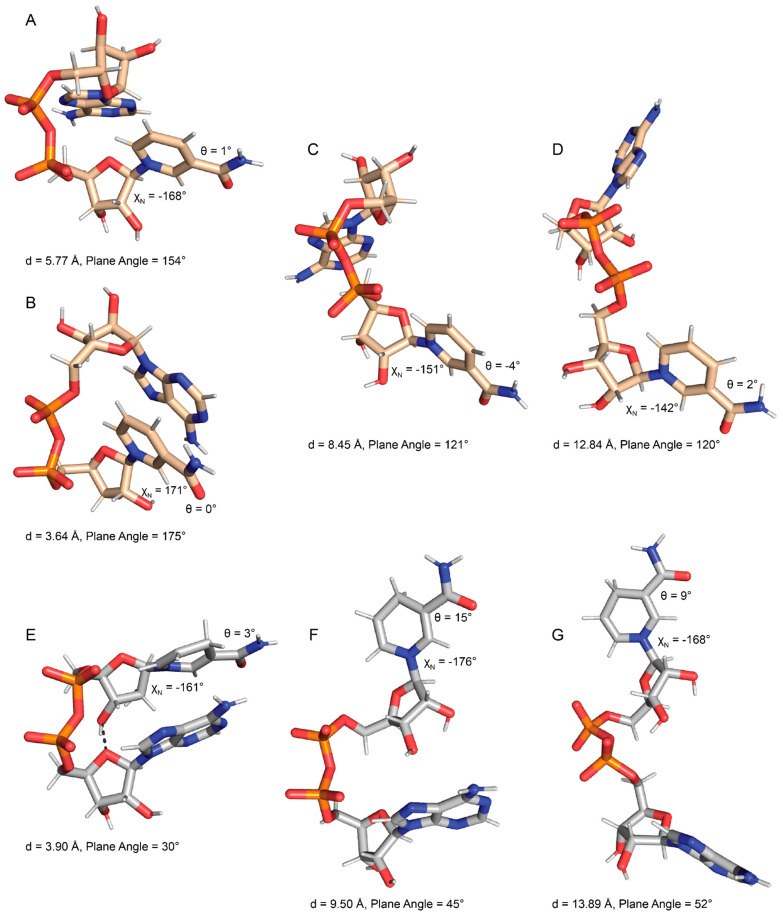
Examples of NAD^+^ (tan) and NADH (gray) conformations. (**A**) The most common folded NAD^+^. (**B**) The less common folded conformation of NAD^+^. (**C**) Semi-extended NAD^+^. (**D**) Extended NAD^+^. (**E**) Folded NADH. (**F**) Semi-extended NADH. (**G**) Extended conformation of NADH. Values of the inter-base centroid distance “d”, plane angle, θ, and χ_N_ are indicated for each pose. Hydrogen bonds are indicated by dashed black lines between heavy atoms. Cluster representations of these conformations are provided in Appendix A.

**Figure 5 ijms-23-11866-f005:**
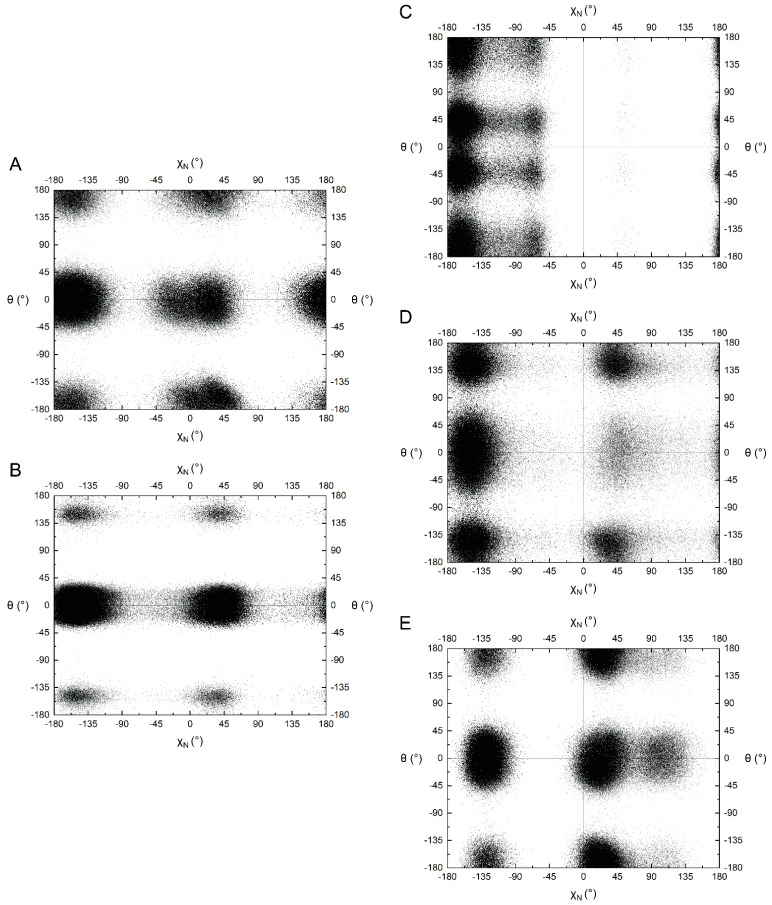
Scatter plot of the nicotinamide/pyridone amide dihedral angle (θ) versus nicotinamide glycosidic bond dihedral angle (χ_N_) for (**A**) NAD^+^, (**B**) NADH, (**C**) 2-ox-NAD, (**D**) 4-ox-NAD, and (**E**) 6-ox-NAD.

**Figure 6 ijms-23-11866-f006:**
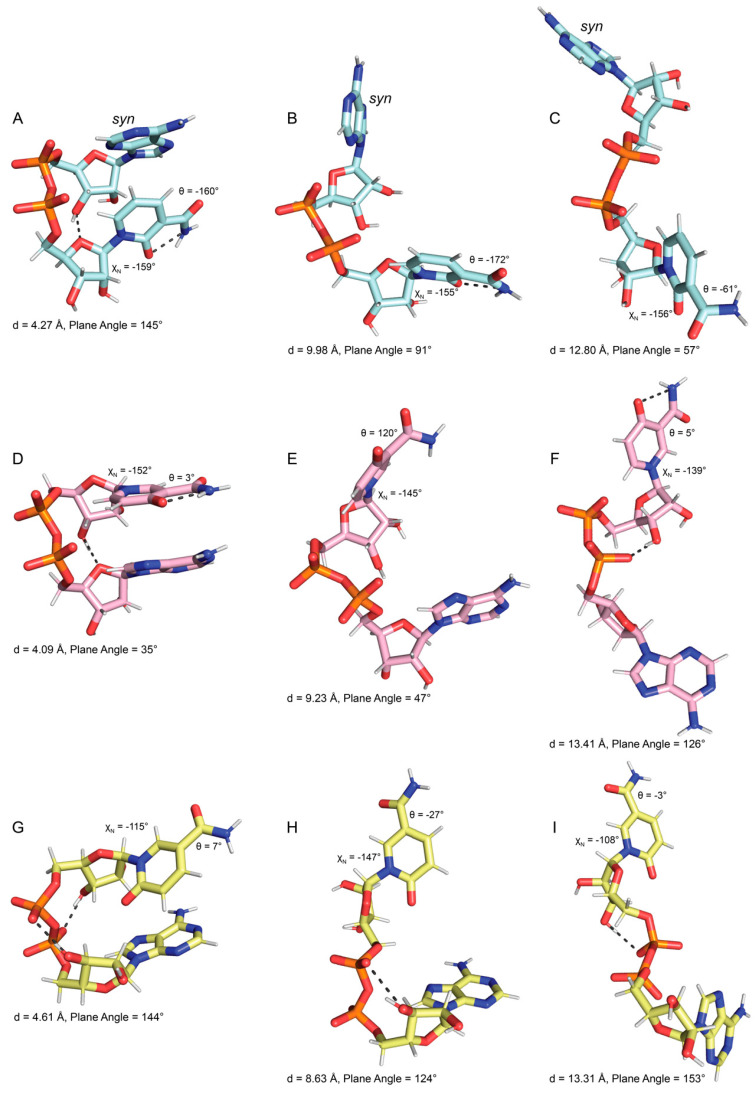
Examples of the most common conformations of ox-NADs. (**A**–**C**) Folded, semi-extended, and extended conformations of 2-ox-NAD (cyan). (**D**–**F**) Folded, semi-extended, and extended conformations of 4-ox-NAD (pink). (**G**–**I**) Folded, semi-extended, and extended conformations of 6-ox-NAD (yellow). Values of the inter-base centroid distance “d”, plane angle, θ, and χ_N_ are indicated for each pose. Hydrogen bonds are indicated by dashed black lines between heavy atoms. All the glycosidic bonds have *anti* conformations except as noted for adenosine in panels A, B, and C. Cluster representations of these conformations are provided in Appendix A.

**Figure 7 ijms-23-11866-f007:**
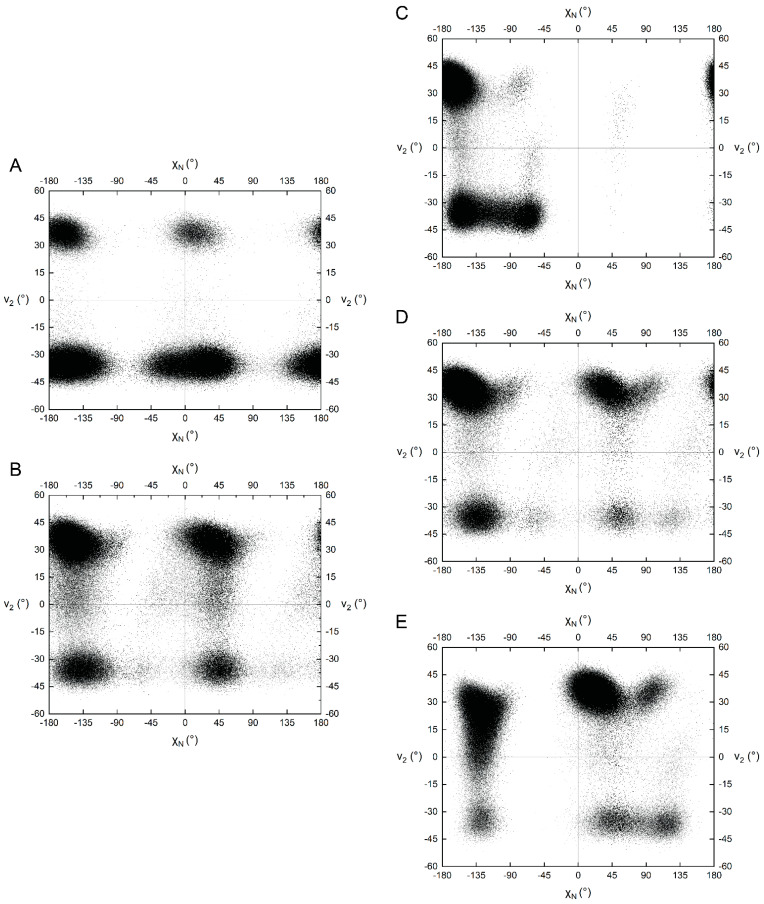
Scatter plot of the nicotinamide/pyridone ribose dihedral angle (v_2_) versus nicotinamide/pyridone glycosidic bond dihedral angle (χ_N_) for (**A**) NAD^+^, (**B**) NADH, (**C**) 2-ox-NAD, (**D**) 4-ox-NAD, and (**E**) 6-ox-NAD.

**Figure 8 ijms-23-11866-f008:**
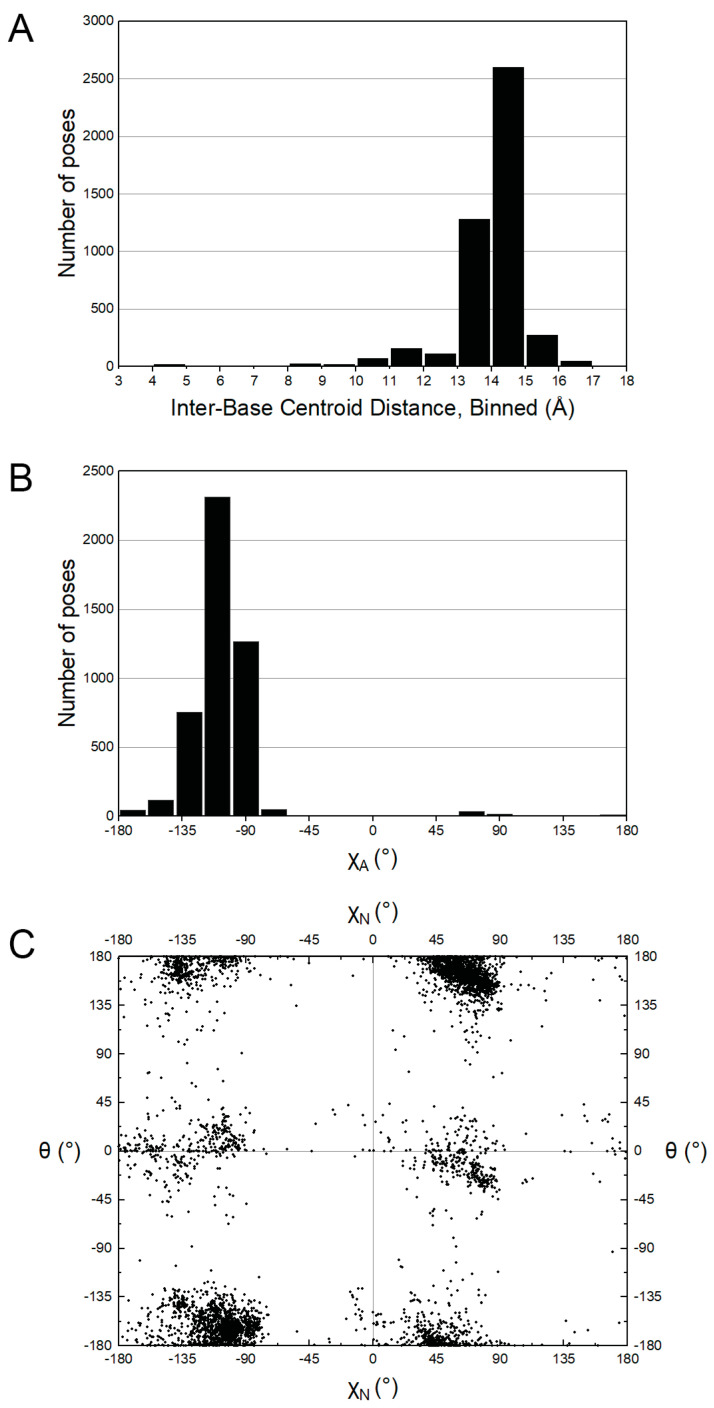
Conformations of protein-bound NAD(H) from PDB. (**A**) Histogram of inter-base centroid–centroid distance. (**B**) Histogram of the adenosine glycosidic bond dihedral angle (χ_A_). (**C**) Scatter plot of the nicotinamide amide dihedral angle (θ) versus nicotinamide ribose glycosidic bond dihedral angle (χ_N_).

**Figure 9 ijms-23-11866-f009:**
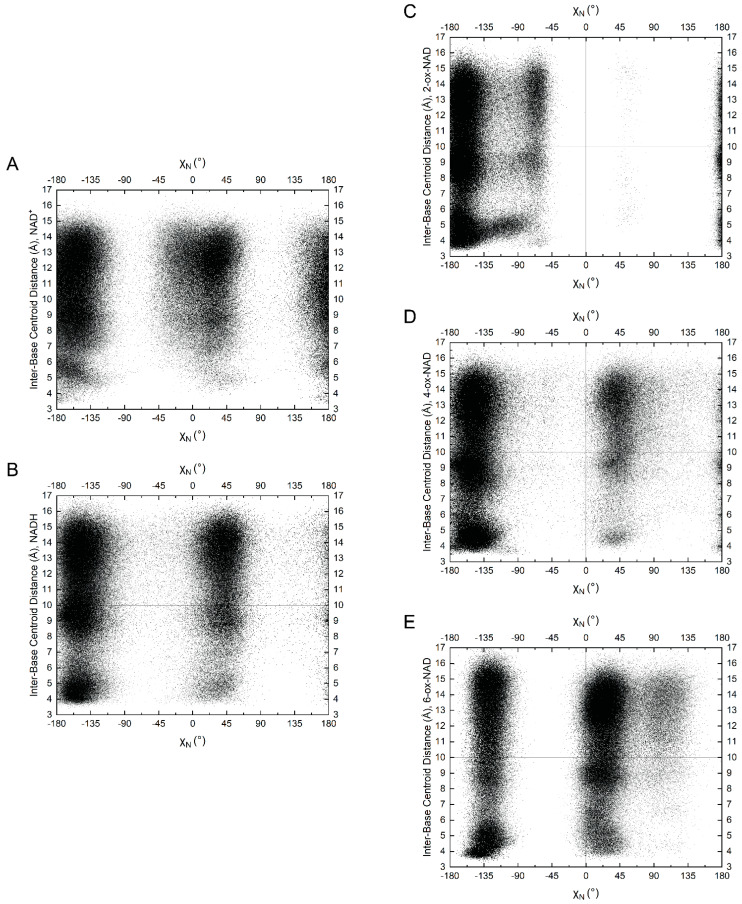
Scatter plot of the inter-base distance versus the nicotinamide/pyridone glycosidic bond dihedral angle (χ_N_) for (**A**) NAD^+^, (**B**) NADH, (**C**) 2-ox-NAD, (**D**) 4-ox-NAD, and (**E**) 6-ox-NAD.

**Table 1 ijms-23-11866-t001:** Dihedral angle definitions.

Dihedral Angle	Adenine (A) or Nicotinamide (N)	Atoms Associated
θ	N	C_4_^N^-C_3_^N^-C_7_^N^-N_7_^N^
ν_2_	N	C_1_′^N^-C_2_′^N^-C_3_′^N^-C_4_′^N^
χ_N_	N	O_4_′^N^-C_1_′^N^-N_1_^N^-C_2_^N^
χ_A_	A	C_4_^A^-N_9_^A^-C_1_′^A^-O_4_′^A^

**Table 2 ijms-23-11866-t002:** Details of the MD simulations.

	NAD^+^	NADH	2-ox-NAD	4-ox-NAD	6-ox-NAD
Simulation time per simulation (ns)	100	100	100	100	100
Number of simulations	10	10	10	10	10
Time step (fs)	2	2	2	2	2
Number of dinucleotide atoms	70	71	70	70	70
Number of water molecules	1221	1115	1227	1234	1220
Number of Na+ ions	1	2	2	2	2
Total number of atoms in system	3734	3418	3753	3774	3732
Ensemble	NPT	NPT	NPT	NPT	NPT
Pressure (bar)	1	1	1	1	1
Temperature (K)	300	300	300	300	300
Box size (nm^3^)	40.64	35.48	40.93	41.69	40.51

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
