# Peer review of "Conformational Preferences of Pyridone Adenine Dinucleotides from Molecular Dynamics Simulations"

_ijms, 2022, doi:10.3390/ijms231911866_

Round 1

Reviewer 1 Report

In the manuscript "Conformational preferences of pyridone adenine dinucleotides from Molecular Dynamics Simulations",
Buckley et al. investigate the conformations sampled by NAD+, NADH and ox-NADs (pyridone dinucleotides, i.e.
oxidized derivatives of NAD with a carbonyl group at the C2, C4 or C6 position)
in solution via Molecular Dynamics (MD) simulations.

The authors analyze preferences for adopting the anti or syn conformations of the pyridone ribose
and also distinguish folded and extended conformations.
One main finding is that 6-ox-NAD, in comparison to the 2- and 4-ox versions
is more extended.
The authors also perform a comparison to structural data contained in the PDB.
Their findings may be of relevance to understand the potential
role of pyridone adenine dinucleotides as enzyme inhibitors
because the binding free energy of the compounds to enzymes is crucially dependent
on a possible conformational change (in addition to desolvation effects).

The manuscript is of interest to the audience of the Int. J. Mol. Sci.
But prior to publishing I have a few things to recommend to strengthen
the impact of the manuscript.

* In the introduction, the authors focus a lot on the biological role of NAD and NADP as well as the oxidized derivatives.
 The authors should, in addition, provide insight into previous simulation as well as experimental
studies (especially NMR, but also e.g. CD) on the conformational preferences of the (ox)-NAD(P) compounds in solution.
 Concerning simulation studies, they should thoroughly discuss potential force-field and sampling issues when it comes
to sampling the conformational equilibria in solution. What kind of force fields have been validated to properly
represent (ox)-NAD(P) compounds? Are there sampling problems?

*  Why do the authors not perform a conformational cluster analysis to characterize the conformations? E.g. cluster
in the space of the relevant internal coordinates. For Fig. 5, one could then plot cluster central conformations?

* It would be nice if the authors provided a critical statement about the extent of conformational sampling done.
  I was unable to open the SupplMat document. Please state in the main article, that the 10 individual simulations
  you performed adequately sampled phase space in case they do so.... If not, consider/discuss enhanced sampling options.

* In the year 2022, there is ample compute power, as the authors point out in comparison to the study from 1999 (Ref. 40).
  However, there has also been a lot of progress in simulation techniques.
  In view of this, I think it is not fair to the importance of the topic of the manuscript to merely extend simulation time.
  One can also perform free-energy calculations, e.g. PMFs between intramolecularly stacked/non-stacked conformations.
  Or, one can simulate some of the protein-bound NAD(H) complexes from the PDB for some hundred nanoseconds.
  The latter is especially important considering that x-ray structures refer to 0 K and do not represent a conformational ensemble, which
  the authors know is important to consider when looking at their compounds of interest. Please comment.

* The diphosphate bridge can adopt different conformations in the protein. What is the relevance of this compared
  to your findings in solution? How would a protein-induced diphosphate-bridge deformation affect the intramolecular
  stacking propensity of the dinucleotides? Please comment.

* The simulation boxes appear very small. In 2022, using the GROMACS simulation engine, one can easily afford somewhat
  larger boxes, which may be important in view of artificial periodicity effects with your charged molecules. Please comment.
  Are conformational preferences the same when simulating in larger boxes?
  Related to this: You discuss that counterion binding is a rare event and that the presence of the counterions has essentially
  no influence on the conformational preferences. Did you do simulations without counterions to verify?

* In the figures, you always specify angles in degrees. Please indicate so in the axes labels for the sake of thoroughness.

Author Response

See attached PDF with replies to the Editor's comments and those of Reviewer 1.

Reviewer 2 Report

In this manuscript the authors studied and compared the conformational preferences of NADH, NAD+ and other oxidated species by using classical MD simulations.

However, some major points need to be addressed prior to publication.

It is known that the parameters of a force field affect the sampling behavior of a molecule and the precision and accuracy of the MD simulations are highly dependent on the reliability of the chosen force fields. Most of the FFs require a manual correction of the force field parameters, especially torsions, using ab initio calculations. (10.3389/fmolb.2021.760283)

Furthermore, if not validated by experimental data, like NMR or  X-ray structures, in silico results remains too speculative and lack of robustness.

The authors applied the CHARMM General Force Field to study NAD molecules in solutions. However, for the well-known NAD+ and NADH compounds, they should perform QM calculations to determine if the geometry and conformational energies are satisfactory and also support MD results with experimental data. Unfortunately, I could not revised to the Supplementary Material (not present in the download section of the on line form)

Here I report the major points that need to be fixed:

·      During the runs of 100ns the NAD(H) alternates between folded and an extended conformations, with a preference for the extended state, is there any experimental data that support this result?

·      How does the force field behave in terms of Energy profiles for the selected dihedral angles of Table 2 with respect to ab initio values? For instance, since the energy barriers χN is affected by the oxidation at C2 and C6, the authors should carefully check the ab initio profile for this torsion. Moreover, the experimental values for NAD(H) should be reported and compared to the in silico results

·      The behavior of the amide dihedral angle (θ) in the free state differs from the values observed for the bound X-ray structures. This effect could depend on the force field parametrization and the authors should compare force field energy profile to  ab initio values to exclude a potential bias in the calculations. How high is the energy barrier for this torsion? Is there an X-ray structure of NAD(H) in the free state?

·      A comparison between the in silico results for the glycosidic bond dihedral angle (χA) and the experimental X-ray data is lacking. Do the values agree?

·       How well does the in silico results reproduce the X-ray structure of NAD(H)? (i.e. RMSD using the starting geometry of MD simulations and other X-ray structures as reference)

Minor revisions

1)    In Fig1A please check the atoms labels with the HB definition in the text and mark with labels the torsions of Table 3 as done for the chemical structure reported in ref 40 (Fig 1)

2)    In Fig7 please add the conformational anti/syn labels under the 3D structures

Author Response

See attached PDF with replies to Reviewer 2.

Round 2

Reviewer 1 Report

The authors have carefully revised their manuscript according to the reviewers' suggestions. I consider the content of the manuscript now of sufficient quality for publication in the IJMS. The authors can still improve (i.e. increase) the size and resolution of axes labels and other labels in figures.

Reviewer 2 Report

Accept in the present form